# Continuous Cold Flow Device Following Total Knee Arthroplasty: Myths and Reality

**DOI:** 10.3390/medicina58111537

**Published:** 2022-10-27

**Authors:** Michele Coviello, Antonella Abate, Francesco Ippolito, Vittorio Nappi, Roberto Maddalena, Giuseppe Maccagnano, Giovanni Noia, Vincenzo Caiaffa

**Affiliations:** 1Orthopaedic and Trauma Unit, Department of Basic Medical Sciences, Neurscience and Sense Organs, School of Medicine, AOU Consorziale Policlinico, University of Bari “Aldo Moro”, Piazza Giulio Cesare 11, 70124 Bari, Italy; 2Orthopaedic and Traumatology Unit, “Di Venere” Hospital, Via Ospedale di Venere, 1, 70131 Bari, Italy; 3Orthopaedics Unit, Department of Clinical and Experimental Medicine, Faculty of Medicine and Surgery, University of Foggia, Policlinico Riuniti di Foggia, 71122 Foggia, Italy

**Keywords:** continuous cold flow therapy, cryotherapy, total knee arthroplasty, pain, opioids consumption, patient satisfaction

## Abstract

*Background and Objectives*: To assess the effect of continuous cold flow (CCF) therapy on pain reduction, opioid consumption, fast recovery, less perioperative bleeding and patient satisfaction in patients undergoing a total knee arthroplasty. *Materials and Methods*: Patients affected by knee osteoarthritis between September 2020 and February 2022 were enrolled in this case-control study. Patients were randomly divided into two groups (*n* = 50, each): the study group received postoperative CCF therapy while the control group was treated by cold pack (gel ice). The CCF device is a computer-assisted therapy with continuous cold fluid, allowing a selective distribution, constant and uniform, of cold or hot on the areas to be treated. In both groups, pre- and postoperative evaluations at 6, 24, 72 h and at the fifth day were conducted using Visual Analogic Scale (VAS), opioid consumption, passive range of motion, preoperative hematocrit, total blood loss by Gross formula, transfusion requirement and patient satisfaction questionnaire. *Results*: One hundred patients, 52 women (52%), were included in the study. Reduction of pain, opioid consumption and increase in passive range of movement were statistically significantly demonstrated in the study group on the first and third days. Patients were satisfied with adequate postoperative pain management due to CCF therapy (*p* = 0.01) and they would recommend this treatment to others (*p* = 0.01). *Conclusions*: A continuous cold flow device in the acute postoperative setting after total knee arthroplasty is associated with pain reduction and improving early movement. Patients were almost satisfied with the procedure. The management of perioperative pain control could improve participation in the early rehabilitation program as demonstrated by the increase in ROM, psychological satisfaction and reduction in opioid use.

## 1. Introduction

Total knee arthroplasty (TKA) is a widely accepted and successful procedure for end-stage arthritis. However, being a major orthopedic intervention, it is accompanied by tissue damage and an inflammatory response, manifesting as local swelling and edema [1,2], reduced range of motion (ROM), stiffness and reduced quadriceps strength [3,4]. Acute postoperative period after TKA may be quite challenging; patients may experience pain, swelling, restricted knee joint excursion, nausea or vomiting, and potential blood loss. A painful acute postoperative period may affect the so called fast track recovery [3], with patient discomfort and long-term effects on knee function [5,6].

Pain is often severe in the early postoperative period, impeding rehabilitation [7]. Multimodal approaches to obtain pain control, throughout advances in anesthetic technique and narcotics administration, is not always sufficient to promote early rehabilitation and is often associated with side effects, such as nausea, vomiting, sedation, pruritus, hypotension, and respiratory depression, which can limit activity, result in longer hospital stays, increase morbidity, and reduce patients’ satisfaction [8,9].

Cold therapy for pain reduction is an accepted and frequently used treatment in daily practice after trauma or surgery. It involves the application of cold to the skin surrounding the injured soft tissues and in joint surgery is supposed to reduce the intraarticular temperature [10] (Figure 1). It is supposed to reduce the local blood flow by vasoconstriction, swelling, and pain experience by slowing the conduction of nerve signals [11,12].

Several cold therapeutic options are available, including first generation cold therapy, such as crushed ice in a plastic bag; second generation cold therapy with circulating ice water with or without compression; and third generation advanced computer-assisted devices with continuous controlled cold therapy. The advantage of these latter devices is the possibility of temperature modulation with cooling at a specific and continuous temperature (generally 5–11°) for a prolonged time. These devices also allow a progressive increase in the temperature before stopping the treatment to avoid secondary cold-induced vasodilatation, and they are not needed to be periodically filled with cold water or ice [13].

Even though some studies show excellent results regarding computer-assisted continuous cold flow therapy, the quality of the available literature is not convincing and level I evidence is still missing [14].

The main aim of this study is to assess the effect of continuous cold flow therapy on pain reduction, opioid consumption, ROM recovery, less perioperative bleeding and patient satisfaction in patients undergoing a TKA.

## 2. Materials and Methods

This is a prospective, case-control, monocenter study, validated by the Ethics Committee (protocol number: 10/CE/2020—1 June 2020) and performed in accordance with the ethical standards laid down in the 1964 Declaration of Helsinki. All the patients involved in the study gave their informed consent prior to their inclusion in the study. 

The patients who were scheduled to undergo primary TKA replacement for grade IV gonarthrosis due to a proven painful knee osteoarthritis, who need to obtain pain relief and improve function, who were able and willing to follow instructions, were enrolled and treated at the Di Venere Hospital of Bari between September 2020 and February 2022.

The patients were divided into two groups using a predefined program (http://www.randomization.com, accessed on 1 September 2020). After surgery, the circulating nurse reviewed the random numbers list. The study group was represented by patients who were treated with postoperative continuous cold flow (CCF) therapy; the control group included patients treated with a cold pack (gel ice) postoperatively.

The inclusion criteria were primary knee osteoarthritis, age between 40 and 81, body mass index (BMI) between 20 and 29.9 and chronic history (for at least 4 months) of knee joint pain.

The exclusion criteria were inflammatory diseases, infection, coagulopathy, previous knee surgery, history of deep vein thrombosis or pulmonary embolism, rheumatoid arthritis, pregnancy and patients who were not able to understand and complete the procedure due to cognitive dysfunction or language barrier.

The sample size as a prospective study was calculated using a pain score tested by visual analogic scale (VAS), referring to previous paper values on the same treatment protocol [15] as primary endpoint. Using a standard deviation of 0.6-point [16,17], we estimated that we would need 23 participants in each group to detect a statistically significant differences at an α level of 0.05 and power level of 80%.

Between September 2020 and February 2022, one hundred and twelve patients with grade IV gonarthrosis were evaluated for eligibility for the present study. Four patients had previous surgery of the knee, four patients participated in another study, two were living in another country, one declined to participate, and one other patient had reprieved the operation. Therefore, they were excluded from the study. Subsequently, 100 patients were included. An overview of the number of patients recruited, enrolled and analyzed in this work is presented in Figure 2 in the manner recommended by consolidated standards of the Strengthening the Reporting of Observational Studies in Epidemiology (STROBE) guidelines [18].

For each patient, the following data were recorded: age; sex; BMI; side of surgery; American Society of Anesthesiologists score (ASA) [19], surgical time; VAS (visual analogic scale) (X), opioid consumption; passive range of motion (ROM); preoperative hematocrit; total blood loss (using the formula described by Gross [7], transfusion requirement and patient satisfaction questionnaire. 

Data were recorded at the following times: T0 (before the surgical procedure); T1 (six-hours post-surgery); T2 (24 h post-surgery); T3 (72 h post-surgery) and T4 (fifth day post-surgery).

Opioid consumption was measured as number of tablets, tramadol (50 mg), used in the postoperative days. All patients received standard pain therapy with acetaminophen 1 g every 12 h, ketorolac tromethamine 30 mg and daily low molecular heparin injections (Fondaparinux, Arixtra, Glaxo-SmithKline, Brentford, Middlesex, UK) as thrombosis prophylaxis up to 4 weeks after surgery.

Passive range of motion as functional recovery was measured by the same operator for each patient using an orthopedic protractor.

Blood loss was evaluated only at T2 using the formula described by Gross.
$$\mathrm{Total}\;\mathrm{blood}\;\mathrm{volume}=\frac{\mathrm{Preoperative}\;\mathrm{hematocrit}-\;\mathrm{postoperative}\;\mathrm{day}\;1\;\mathrm{hematocrit}}{\mathrm{Average}\;\mathrm{hematocrit}\;\mathrm{between}\;\mathrm{preoperative}\;\mathrm{and}\;\mathrm{postoperative}\;\mathrm{day}\;1}$$

Transfusion requirement was evaluated as the number of blood transfusions during hospitalization (transfusion trigger was set at 8.0 g/dL of hemoglobin).

Patient satisfaction was assessed via questionnaire. Participants reported their overall satisfaction by placing an X on a 10-cm line (from 0 (extremely unsatisfied) to 10 (extremely satisfied)). The distance was measured in centimeters from the beginning of the line to the center of the patient’s X, using a ruler with 1-mm increments. Participants were also asked to recommend the method of cooling, answering “yes” or “not”.

According to a standardized protocol, patients received antibiotic prophylaxis with cefazolin 2 g and premedication 2 h before operation with acetaminophen 1 g. Patients were operated under spinal anesthetic treatment by one experienced knee surgeon (VC), performing a minimum of 150 TKA procedures annually. Patients were operated on using a midline anterior incision with a medial parapatellar arthrotomy and with the use of dedicated instruments for the implantation of a cemented Postero-Stabilized (PS) TKA [20] (Zimmer Biomet, Warsaw, IN, USA). No tourniquet was used during surgery. Before skin closure, hemostasis was controlled using diathermia. The wound margins were infiltrated with 140 cc ropivacaine (0.2%) as local infiltration anesthesia (LIA). No adrenaline was used during LIA, since it was shown that adrenaline could be omitted from the LIA-mixture [21]. A wound drain was used. All wounds were closed with a barbed suture wire. The surgical wound was covered with a hydrocolloid dressing (Aquacel^®^ Surgical, ConvaTec Inc., Reading, UK) for 7 days and a compressive dressing was applied in the operating room. A strict urinary nurse-led bladder scan management protocol was used for the prevention of postoperative urinary retention [22]. All patients were planned to be discharged on the fifth day after surgery. 

The study group received advanced cooling devices. The machine used (Zamar Z-one MG465A, Croatia, Balkans) was dedicated to each patient during the entire period of hospital stay. It is a computer-assisted continuous cold flow therapy allowing a selective distribution, constant and uniform of cold or hot, on the areas to be treated. Patients cooled the affected knee according to a fixed protocol.

The application started the day before surgery for 3 h and repeated within 6 h post-operation for 3 h. Then it was applied for 3 h respectively in the morning and in the afternoon, during the five postoperative days.

Temperature was set at 5° as efficient CCF therapy requires cooling the skin and achieving reduction in pain according to Chesterton et al. [1]. Compression was granted at the minimum level to avoid discomfort.

In the control group patients received 15 min cold pack (conserved at −17°) treatment with two cold packs anterior and one posterior to the knee within 6 h post-operation and then repeated at 2 and 4 h after surgery. The following days patients received the same cold pack 15 min after each physiotherapy session (10 am and 3 pm). Cold packs were also permitted as requested during the night. No compression was applied.

Thereafter the same rehabilitation program was employed for all patients, consisting of full weight bearing and active range of motion exercises at day of surgery. A continuous passive motion device was used twice a day. 

As a primary endpoint, pain was quantified using the VAS scale with scores ranging from 0 (no pain) to 10 (worst imaginable pain).

The functional recovery, pain management, blood loss and patient satisfaction were assessed as a secondary endpoint.

A prospective clinical study was conducted. The data were collected and analyzed using SPSS (v 23; IBM^®^ Inc., Armonk, NY, USA). Descriptive statistics were calculated for the overall sample and for follow-up. Categorical variables were presented as numbers or percentages. Continuous variables were presented as mean and standard deviation. Due to the non-homogeneous distribution of the values using the Kolmogorov–Smirnov test (*p* > 0.05), non-parametric tests were considered. To compare the average values between the groups at the same times, the U Mann–Whitney test or Fischer’s test were used, when appropriate. To compare the value within the same group at different times, the Wilcoxon test or Related-Samples Friedman’s test Two-Way Analysis of Variance were used. To demonstrate the correlation between the CCF device and variables, a multiple regression model was then fitted. For all the tests, a *p*-value of less than 0.05 was considered to be statistically significant.

The data presented in this study are available on request from the corresponding author.

## 3. Results

One hundred consecutive patients treated with TKA were enrolled in this study and allocated into two groups, with 50 cases in each group. We compared the study and control group at recruitment (Table 1). No statistical differences emerged between groups.

None of the patients experienced any skin complications due to the CCF therapy or gel ice. None of patients needed revision surgery due to infection and mechanical relaxation in the early period. Neither intraoperative nor postoperative complication was recorded.

The VAS scores before and after surgery were noted for each time. No difference was shown preoperatively. The study group recorded lower values if compared with control, but only at one day postoperatively was the difference statistically significant (*p* = 0.01, Table 2 and Figure 3).

In the study group a reduction in opioid consumption was demonstrably statistically significant in the first- and third-days postoperative. Overall, there was a lower consumption in the study group than in the control (*p* = 0.02, Table 2).

Descriptive values and comparison results of passive knee range of motion values are given in Table 3. It was found that the difference between preoperative and postoperative knee flexion measurements was significantly improved in both groups (*p* = 0.01 in both groups using Two-Way Analysis of Variance) and the differences between the groups were changed one- and three-day postoperative.

No statistically significant difference was found between the groups regarding blood loss and transfusion number (Table 3).

Patients treated with CCF therapy were more satisfied than the control group (Table 3).

The multiple linear regression models showed postoperative VAS score and opioid consumption were influenced by CCF therapy rather than other preoperative variables (Table 4).

## 4. Discussion

After a major orthopedic surgical procedure, such as TKA, pain control is imperative to promote rapid recovery and collaboration to the rehabilitation program. Nowadays, specific pathways of care, including enhanced recovery after surgery (ERAS), have been introduced. It has been proposed these pathways, along with multimodal analgesia, have reduced the length of hospital stay (LOS) for patients undergoing TKA [23]. Analgesic options for TKA include pre-emptive analgesia, local infiltration, systemic analgesics (opioids, non-opioids, and patient-controlled analgesia), epidural analgesia and, more recently, femoral nerve block [24]. It has been postulated that peripheral nerve blocks provide intense, site-specific analgesia and are associated with a lower incidence of side effects when compared with many other modalities of postoperative analgesia [25]. Another approach to reduce pain and promote recovery consisted of cooling obtained by a different kind of CCF therapy approach [26,27]. However, consensus has not successfully been reached on the better modality, its efficiency and scheme of application.

The most important finding in our study is that postoperative computer assisted CCF therapy is effective in pain control after TKA and associated with reduction in analgesic use during the early postoperative period. It is also associated with a major passive ROM during the first two days and with a great grade of patient satisfaction. We enrolled a larger study population if compared to power analysis due to the possibility of patient refusal and to strengthen the results.

Significant reduction in VAS was observed only at T2 (24 h), however there was a reduction in VAS after every cooling session in the study group. Similar to these findings, pain reductions in the acute postoperative phase were reported [7,28,29]. Inflammatory conditions secondary to surgery cause most of the perceived pain in the immediate postoperative period after knee replacement. Additionally, patients undergoing knee surgery have already suffered pain for a long period secondary to established central nervous system sensitization. It seems to be intuitive that any approach aimed to reduce pain should be pursued to avoid risk of development of chronic pain [30]. Bech et al. found no superiority in reducing pain compared with traditional icing. Cooling reduces the tissue metabolic rate and relieves inflammation by suppressing enzymatic activity and preventing secondary tissue damage, reducing muscle tissue spasm [31].

Consumption of analgesic was also investigated in this study. In our paper the consumption of additional opioids as rescue medication was up to two times greater in the control group than in the study group. However, this trend was significant only in the first two postoperative days, with these days being the worst period for pain management.

The incremented use of opioids can slow a patient’s rehabilitation program due to side effects (nausea, vomiting, headache, constipation). A slower rehabilitation can be associated with retarded discharge and increased costs [32].

Su et al. [33] showed a lower consumption of opioids in a multicenter RCT. In another recent systematic review and meta-analysis of 11 RCT, Addie et al. [28] showed no benefits for analgesic use. Thienpont et al. [14], on the contrary, found no difference in VAS and morphine use in their RCT. They shared Algafly’s conviction that a decreased nerve conduction velocity resulting in better pain tolerance can explain the local anesthetic effect of intense cooling [11]. However, this is merely a temporary measure that disappears after ice removal.

The CCF therapy is crucial and ensures that the knee ROM movements start in the early postoperative period. There were studies reporting that the CCF therapy affects the knee movement gap in a positive way in the early postoperative period [34,35,36]. In our study passive ROM at first and second postoperative day was better in the cold compression group. Su et al. found no differences for ROM or functional testing between a cryopneumatic device and ice packs [33]. Kullemberg et al. [12] and Holmstrom et al. [37] reported that cold compression was more efficient than epidural analgesia with better ROM, reducing hospital stay.

Patients reported high satisfaction rate with postoperative cooling and were likely to recommend icing device. Less pain could substantiate this validation. The high level of satisfaction may also be associated to greater patient contact with the staff conduction device function and periodic skin and dressing check. The psychological aspect plays a key role in rehabilitation [38]. Only two patients reported that the device was uncomfortable and that some nurses appeared not to be sure how to apply it. 

Blood loss evaluation is a key point to justify routine use of a cold flow device, considering its additional costs. Reduction in blood loss and blood transfusion may reduce costs associated with TKA. Additionally, blood transfusion is associated with an increased rate of periprosthetic infection. In our study we found that the total blood loss measured by Gross formula in the postoperative period in the study group was lower than in the control group; however, non-statistically significant values were observed in total blood loss. Kuyucu et al. [26] also did not find significant difference in blood loss between their study groups. Ruffilli et al. [7] carefully evaluated blood loss throughout drain output, blood loss with Gross formula, transfusion requirements, and did not find any difference between their study groups. Contradictory results have been found by several authors regarding visible and real blood loss [37,39]. This could be explained by the fact that CCF therapy may cause an immediate reduction of bleeding by vasoconstriction and compression but later lead to a cold-induced vasodilatation and interfere with secondary hemostasis [40]. Hemostasis and coagulation are biochemical reactions that are most efficient at basal body temperature. Computed control of temperature progressive increase before cryotherapy removal should be considered to reduce secondary vasodilatation and blood loss [41]. Karaduman et al. found the Hb levels measured in the ice pack group were lower than those in experimental groups treated with CCF therapy in the postoperative period. They also sustained that postoperative CCF therapy was effective in reducing blood loss. However, preoperative CCF therapy application had no significant effect [15].

The evaluation method of blood loss could explain the difference in results in the literature. We decided to use the formula described by Gross et al. [42] for blood loss calculation after arthroplasty surgery, which is one of the most popular among surgeons. However, the Gross equation does not involve Hb-related factors. Actual blood loss and anemia are revealed by the calculated perioperative volume and the changing in hematocrit [42]. This could consist of a limitation of this method and additional evaluation of Hb values and drainage blood loss should be considered.

The novelty of this study is the careful analysis of the patient’s outcomes combined with their satisfaction. Psychological well-being improves clinical outcomes and makes the patient prone to better physiotherapy [38].

Our study has several strengths. This was performed in one single center, one single experienced surgeon performed all surgeries using the same surgical approach, one type of anesthesia, implant, pain and rehabilitation program. The self-reported patient satisfaction questionnaire is an additional element considering the importance of a fast-track recovery plan in replacement surgery.

However, some limitations must be considered. The sample size could be increased. Better evaluation of blood loss throughout alternative formula, including Hb values, should be adopted. Another limitation is that patients were evaluated only during the hospital stay so it cannot draw definitive long-term conclusions about CCF therapy efficacy.

## 5. Conclusions

Continuous cold flow devices in the acute postoperative setting after TKA are associated with pain reduction and improving early ROM. Although reduction in blood loss throughout the Gross formula was observed, no significant differences were found. Patients were almost satisfied with the procedure and should recommend it to improve perioperative pain control and participation to the early rehabilitation program due to lower opioid consumption and higher passive ROM.

## Figures and Tables

**Figure 1 medicina-58-01537-f001:**
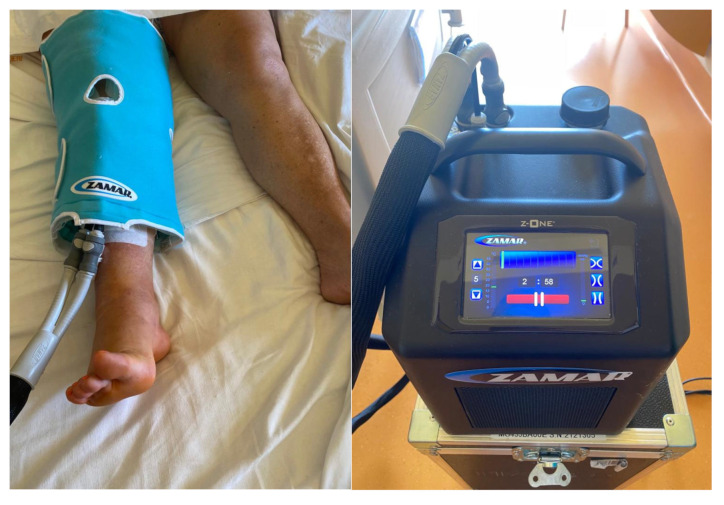
Application of cryotherapy pad on a patient.

**Figure 2 medicina-58-01537-f002:**
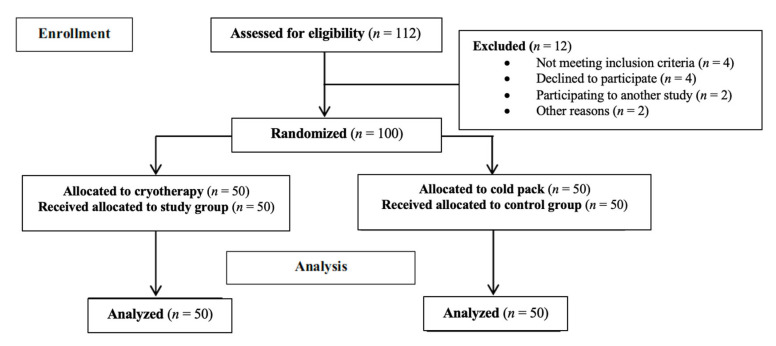
Diagram of the number of patients enrolled and analyzed in this study using STROBE guidelines.

**Figure 3 medicina-58-01537-f003:**
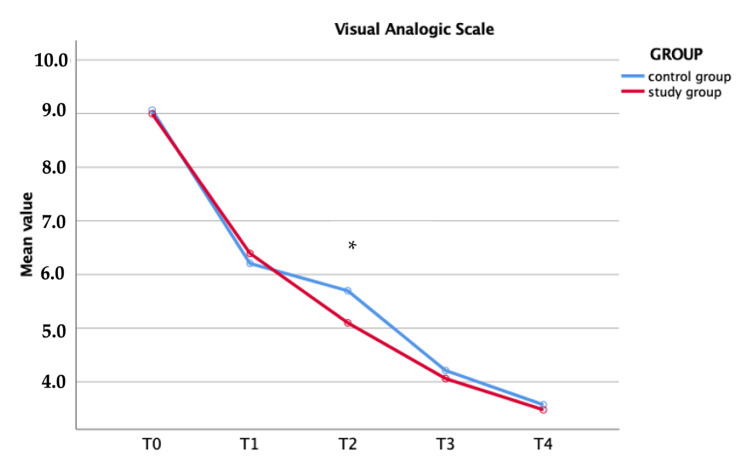
Visual analogic scale within each group (* *p* = 0.01).

**Table 1 medicina-58-01537-t001:** Main data of the study.

Preoperative Features	Study Group	Control Group	*p*-Value
**Age (year)**	66.56 ± 6.78	65.76 ± 6.23	0.63
**Gender (female)**	27 (54%)	25 (50%)	0.84
**BMI (Kg/m^2^)**	28.50 ± 4.32	27.82 ± 3.47	0.56
**Side (left)**	23 (46%)	21 (42%)	0.84
**ASA Classification**			0.86
I	20 (40%)	18 (36%)	
II	16 (32%)	14 (28%)	
III	11 (22%)	13 (26%)	
IV	3 (6%)	5 (10%)	
**Preoperative hematocrit (%)**	38.82 ± 3.67	39.73 ± 3.87	0.16
**Surgical time (min)**	57.09 ± 15.04	58.16 ± 15.46	0.70

(One hundred patients; U Mann–Whitney and Fischer’s test; data are presented as mean ± standard deviation or number and percentage; BMI: Body Mass Index; ASA: American Society of Anesthesiologists score).

**Table 2 medicina-58-01537-t002:** Differences in pain and opioid treatment between groups.

VAS		Study Group	Control Group	*p*-Value
	T0	9.00 ± 0.47	9.06 ± 0.49	0.52
	T1	6.39 ± 1.23	6.20 ± 1.12	0.46
	T2	5.09 ± 0.94	5.69 ± 1.08	**0.01**
	T3	4.06 ± 0.89	4.21 ± 0.72	0.51
	T4	3.48 ± 0.68	3.57 ± 0.64	0.52
**Opioid consumption**				
	T1 (mg)	8 (400)	9 (450)	1.00
	T2 (mg)	17 (850)	32 (1600)	**0.01**
	T3 (mg)	9 (450)	19 (950)	**0.05**
	T4 (mg)	11 (550)	14 (700)	0.64
	Total (mg)	45 (2250)	74 (3700)	**0.02**

(One hundred patients; U Mann–Whitney test for VAS and Fischer’s test for opioid consumption; data are presented as mean ± standard deviation or number of tablets tramadol and equivalent mg; VAS: visual analogic scale).

**Table 3 medicina-58-01537-t003:** Differences in passive range of motion, total blood loss using Gross formula and patient satisfaction between groups.

Passive ROM (°)		Study Group	Control Group	*p*-Value
	T0	84.40 ± 7.14	83.48 ± 7.02	0.55
	T2	111.57 ± 7.04	105.49 ± 11.24	**0.01**
	T3	110.94 ± 7.52	107.39 ± 7.89	**0.01**
	T4	108.84 ± 6.07	108.22 ± 6.61	0.64
**Total blood loss** **(Gross formula)**				
	T2	1.03 ± 0.42	1.06 ± 0.55	0.86
**Number of transfusions**		4 (8%)	5 (10%)	0.50
**Patient satisfaction (cm)**		8.55 ± 0.36	6.05 ± 0.58	**0.01**
**Patient who recommended “yes”**		43 (86%)	31 (62%)	**0.01**

(One hundred patients; U Mann–Whitney test and Fischer’s test; data are presented as mean ± standard deviation or number and percentage; ROM: range of motion).

**Table 4 medicina-58-01537-t004:** Multiple linear regression models for postoperative VAS and opioid consumption.

	VAS				Opioid Consumption			
	B	95% CI		*p*-Value	B	95% CI		*p*-Value
**Intercept**	7.50			<0.01	0.915			0.05
**Group (cryotherapy)**	−0.62	−1.04	−0.20	<0.01	−0.31	−0.50	−0.11	<0.01
**Age**	−0.01	−0.04	0.02	0.57	0.01	−0.01	0.03	0.15
**Sex (female)**	−0.25	−0.93	0.41	0.45	0.12	−0.19	0.44	0.44
**Left knee**	0.53	−0.14	1.20	0.12	−0.21	−0.52	0.10	0.18
**BMI**	−0.01	−0.06	0.05	0.93	−0.02	−0.05	0.01	0.12
**ASA Classification**	−0.01	−0.24	0.22	0.96	−0.07	−0.17	0.04	0.22
**Preoperative** **hematocrit**	−0.01	−0.07	0.04	0.69	−0.01	−0.03	0.02	0.72
**Surgical time**	−0.01	−0.02	0.01	0.32	−0.01	−0.01	0.01	0.69
**Total blood loss**	−0.20	−0.63	0.25	0.39	−0.01	−0.21	0.20	0.99

(BMI: Body Mass Index).

## Data Availability

The data presented in this study are available on request from the corresponding author.

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
