# Peer review of "Continuous Cold Flow Device Following Total Knee Arthroplasty: Myths and Reality"

_medicina, 2022, doi:10.3390/medicina58111537_

Round 1
Reviewer 1 Report
Cryotherapy is a broad subject, and in the title it should be clearer that the paper focuses on joint cooling, as opposed to e.g. nerve freezing to prevent postoperative pain.
Anaemia seems to be an incidental occurrence and observation in this instance. There is no immediate association to the proposed cryotherapy, and it's effects from the therapy are entirely obscure. The link to cryotherapy is not offered an explanation, either. Perhaps it should be disregarded altogether. Certainly, the paper does not benefit from how this theoretical result might have been obtained, and the result may be entirely incidental. It leaves the reader bemused. It also leaves the impression that the authors have tried to beef up the results, by claiming it is an outcome resultant from the intervention, which it is not.
PROM - is a term usually reserved as an abbreviation for 'Patient-Reported Outcome Measures', please use another term.
The significance of the outcome may have been markedly improved if there was a much longer follow-up period, e.g 6 or 12 months. The success of TKA surgery often hinges on the ability to control immediate post-operative pain, and not just provide short-term comfort and pain relief. If the authors had longer follow up periods, then cryotherapy could provide useful improvements in TKA success rates, which is warranted.
Along those lines, the use of ultra-sound guided nerve blocks should be discussed as a means to improving both short term analgesia but also long term TKA successes.
The main outcome remains a small, if significant statistically, reduction in NRS/VAS of less than 1 at a single time point postoperatively. This is a clinically underwhelming finding. Pt. acceptance is all very well, but the scientific novelty is limited, as is probably the clinical impact of the study. Without the very tentative suggestion that there might be an unexplained association to anemia, this is a very slim paper, which may be difficult to publish in a medical journal.
Reviewer 2 Report
The contribution of a low cost method with wide benefits is important.
The results obtained have followed a correct methodology.
I think it has enough quality to be published
Reviewer 3 Report
Finding ways to minimize the pain and optimize the recovery after total knee arthroplasty is relevant. However, the authors should consider the following comments to improve their manuscript.
Using the term "cryotherapy" is confusing here as the auhors used it for qualifying the continuous cold flow device they monitored through computer and not the control procedure using cold packs -which is also "cryotherapy".
Another confusing part of the manuscript is the difference in applying the cooling in the experimental setting and in the control procedure. One could argue that with changing the patterns of appying ice packs, one could also get better benefits, and similar benefits as those obtained with the tested device.
The statistics should be reviewed, maybe with the help of a statistician; especially for calculating the number of patients to get a better outcome of one procedure versus another.
Abstract : Maybe the authors should consider re-writting ther abstract. It is difficult for the reader to understand what are the differences between “postoperative cryotherapy” and the treatment by cold pack (gel ice). Some words of explanation are needed.
? pre and maybe also post-operative hematocrit,?
… “One-hundred patients, more than half women, were included in the study”: already mentioned. Mention the exact number of patients right after “Material and methods
“Total blood loss was less in the study group but not statistically significant (p=0.86)”: not significant, do not mention in the abstract
“Continuous cold flow device in the acute postoperative setting after total knee arthroplasty”: this is I guess what you designed as cryotherapy. I would recommend you do not use cryotherapy, as application of cold packs can also be named as cryotherapy.
Express that two kinds of cooling devices have been studied
Patients were almost satisfied with the procedure: not needed
“The management of perioperative pain control could improve participation to the early rehabilitation program.”: not clear
Intro
A a painful acute postoperative ?? delete the extra a
Material and methods:
The authors expressed: “Sample size was calculated using pain score tested by Visual Analogic Scale (VAS) as primary endpoint. Using a standard deviation of 0.6-point [15], we estimated that we would need 23 participants in each group to detect a statistically significant differences at an α level of 0.05 and power level of 80%.” Actually, what the authors need to calculate is the number of patients to see a superiority of the examined device compared to the control.
The authors nevertheless choose to investigate much more patients than what was aimed. Maybe the number they choose is compatible with a superiority study design.
End page 3: why is post-operative haematocrit not mentioned?
Not clear why the procedure was not similar with the cold packs and the continuous cold flow device in terms of frequency of application
Discussion
mets-analysis => meta-analysis
Conclusion:
The part “Although reduction in blood loss throughout postoperative hematocrit levels was observed, no significative differences were found. Patients were almost satisfied with the procedure and should recommend it to improve perioperative pain control and participation to the early rehabilitation program.” does not really fit and is not relevant in my opinion
Round 2
Reviewer 1 Report
Good recovery. Sufficient to make it publishable.
Should go straight to proof from here.
No further comments